# Towards Innovative Solutions for Monitoring Precipitation in Poorly Instrumented Regions: Real-Time System for Collecting Power Levels of Microwave Links of Mobile Phone Operators for Rainfall Quantification in Burkina Faso

Moumouni Djibo [1,2], Wend Yam Serge Boris Ouedraogo [1], Ali Doumounia [1,3], Serge Roland Sanou [1,4], Moumouni Sawadogo [1], Idrissa Guira [5], Nicolas Koné [6], Christian Chwala [2,7], Harald Kunstmann [2,7] and François Zougmoré [1,*]

1  Laboratoire de Matériaux et Environnement (LA.M.E), Université Joseph KI-ZERBO (UJKZ), Ouagadougou 03 BP 7021, Burkina Faso
2  Institute of Geography, University of Augsburg, 86159 Augsburg, Germany
3  Institut Des Sciences (IDS), Ouagadougou 01 BP 1757, Burkina Faso
4  Autorité de Régulation des Communications Électronique des Postes (ARCEP), Ouagadougou 01 BP 6437, Burkina Faso
5  Telecel Faso SA, Ouagadougou 08 BP 11059, Burkina Faso
6  Institut de l'Environnement et de Recherches Agricoles (INERA/CNRST), Ouagadougou 03 BP 7047, Burkina Faso
7  Institute of Meteorology and Climate Research, Karlsruhe Institute of Technology, Campus Alpin, 82467 Garmisch-Partenkirchen, Germany
*  Correspondence: zougmore@ujkz.bf

**Abstract:** Since the 1990s, mobile telecommunication networks have gradually become denser around the world. Nowadays, large parts of their backhaul network consist of commercial microwave links (CMLs). Since CML signals are attenuated by rainfall, the exploitation of records of this attenuation is an innovative and an inexpensive solution for precipitation monitoring purposes. Performance data from mobile operators' networks are crucial for the implementation of this technology. Therefore, a real-time system for collecting and storing CML power levels from the mobile phone operator "Telecel Faso" in Burkina Faso has been implemented. This new acquisition system, which uses the Simple Network Management Protocol (SNMP), can simultaneously record the transmitted and received power levels from all the CMLs to which it has access, with a time resolution of one minute. Installed at "Laboratoire des Matériaux et Environnement de l'Université Joseph KI-ZERBO (Burkina Faso)", this acquisition system is dynamic and has gradually grown from eight, in 2019, to more than 1000 radio links of Telecel Faso's network in 2021. The system covers the capital Ouagadougou and the main cities of Burkina Faso (Bobo Dioulasso, Ouahigouya, Koudougou, and Kaya) as well as the axes connecting Ouagadougou to these cities.

**Keywords:** commercial microwave links; power level; SNMP protocol; acquisition system; rainfall estimation

## 1. Introduction

### 1.1. Background and Rationale

Today, global warming is one of the most worrying threats to the future of mankind. Indeed, at the global level, the warming of the atmosphere and oceans, the melting of glaciers and snow cover, and the rise in sea level are observed. West African Sahel is experiencing changes in the distribution of precipitation over time and an increase in the occurrence of extreme events, such as heatwaves and floods [1]. The majority of Sub-Saharan Africa countries' land area for agriculture and approximately 97% of crop

production in these countries is rain-fed [2]. In Burkina Faso, for example, over the 40 years between 1960 and 2000, cereals production almost exactly followed variations in rainfall [3].

Most of the water in these countries comes from rainfall. In Burkina Faso, for example, more than 90% of the water comes from atmospheric precipitation. Therefore, rainfall monitoring is of paramount and increasing importance for agriculture, water resources management, floods, etc. However, in the sub-Saharan region, the availability of measuring instruments is limited and deteriorating. This limits the knowledge about rainfall distribution, creates significant risk and uncertainty, and prevents the agricultural sector from adopting appropriate measures. Improved control and management of water resources is, however, vital for these countries' development. Therefore, it is essential to significantly improve the knowledge of spatial and temporal rainfall distribution. A new, reliable, and accessible method, also for countries in Africa, Latin America, and South-East Asia, is therefore welcome. Within the TOP-RainCell project we have thus set out to acquire and process data from commercial microwave links (CMLs) for country-wide rainfall estimation in Burkina Faso.

### 1.2. Rainfall Estimation Methods

For the study of precipitation, in particular rainfall quantification, several methods are used, e.g., rain gauges, ground-based radar, and satellites [4].

Traditionally, the most used instrument for measuring rainfall on the ground is the rain gauge. This technique has the disadvantage of low spatial representativeness. In addition, rain gauges require regular maintenance visits. Furthermore, there is often a significant delay in the rain gauge data delivery for post-processing, e.g., at the Global Precipitation Climatology Centre, leading to low sensor density in station-based rainfall products [5].

Weather radar, applied to rainfall estimation, appears to be more accurate in time and space than rain gauges [6]. It allows for the determining of the position, intensity, and displacement of the rain event, within a coverage radius typically between 100 km and 150 km [6]. The weather radar is a high-tech device that requires significant financial investment for its acquisition and human resources of high technical expertise for operation and maintenance. In addition, one needs several radars in order to cover the territory of a given country. For a country such as Burkina Faso, with an area of 274.200 km$^2$, nine weather radars with a range of 100 km would be needed.

Satellite-based remote sensing is another important technique for monitoring rainfall [7,8]. The advantage of remote sensing with low earth orbit satellites is the very good global, or almost global, coverage—depending on the orbit–spatial coverage. However, satellite data are subject to uncertainties and have low temporal resolution for continuously estimating the precipitation at a given location [7,8].

The use of CMLs to quantify precipitation is an alternative that is increasingly being explored by many researchers [9–12]. When a microwave signal propagates between a transmitting and a receiving antenna, its signal strength decreases due to free space loss [13], atmospheric gases, and associated effects [14]. If the signal passes through rainfall, it is attenuated significantly due to the phenomena of absorption and scattering of electromagnetic radiation by water drops. This additional attenuation can be used to determine the amount of rainwater falling between the two antennas [10,15,16].

Using the CML technology to quantify rainwater has the following advantages:

(i)    it is possible to cover very large areas;
(ii)    there is no need for major new investments;
(iii)    it is fully automatic and does not require experienced service personnel to operate it;
(iv)    there is the possibility of having the data in real time.

Another advantage of the CML data technique is that the quantifications take place at a maximum altitude of 100 m, above the ground, unlike other remote sensing techniques, such as radar or satellites, which scan precipitation at higher altitudes. There is therefore more precision at small spatial and temporal scales with this technique.

This technology using telecommunication commercial microwave link networks for precipitation quantification was first initiated and implemented in Israel 2006 [9]. The correlation between precipitation intensity measured by the CML and the rain gauge was 0.86 for a 15 min interval and 0.9 for an hourly interval. In 2008, in Israel, by using a great number of CMLs, Zinevich et al. [17] inferred the spatial evolution of precipitation. Subsequently, it was applied in several other countries:

- Netherlands: in 2007, Leijnse et al. [18] performed the first analysis of data from two CMLs in the Netherlands. Eight rainfall events were evaluated over two CMLs with a resolution of 15 min, and the results are consistent with the rainfall retrieved by the rain gauges and the C-band radar. In 2013, Overeem et al. (2013), by increasing the number of CMLs, were able to deduce the spatial evolution of rainfall.
- Germany: in 2012, Chwala et al. [19] processed the first CML data that they recorded first using data loggers. A new algorithm based on short-time Fourier transform (STFT) was proposed for wet/dry classification on five CML links. The correlation reached 0.81 for the link gauge comparison.
- Burkina Faso: less than 6 years after its first implementation, this new technology was initiated and tested for the first time in Africa. The implementation was performed during the monsoon season of 2012–2013, between two towers 29 km apart, between Korsimoro and Kaya, two cities of Burkina Faso. The rainfall amount during this period were quantified and compared with conventional rain gauge measurements. The experiment was highly successful, and the effectiveness of the method was demonstrated [16].

Due to the sparseness of weather station networks and the lack of weather radars in developing countries, the additional precipitation information derived from the CML is particularly important in African, Southeast Asian, and Latin American countries.

When the TOP-RainCell technology was first implemented in 2012, the data were stored at the mobile phone operator's site and periodically retrieved on a USB stick and transferred to the laboratory. As this was the early days of this technology, this was how performance data was collected for rainfall quantifications [17,19,20]. This data acquisition approach may result in limited or irregular access to the data.

As time went on, relations with the operators became stronger and more trusting, and the use of a significant number of microwave links began to be considered. Likewise, the collection technique and quantification methods improved with a growing number of laboratories around the world [10,12,20–22] and TOP-RainCell in Burkina Faso.

Several teams have set up continuous data acquisition systems, for collecting CML power levels, either serially [21] or in parallel [22].

In 2015, Fencl et al. [21] proposed a custom data acquisition software running within an operator's CML network. This software is only capable of serial data acquisition and does not appear to be suitable to provide data for a large number of CMLs with consistently high temporal resolution.

In Germany [22], a parallel data collection system that allows simultaneously access and record data of several CMLs has been implemented. However, the data are stored by the mobile phone operator and researchers use a Virtual Private Network (VPN) link from the laboratory to gain access to the data server located in the cell phone operator site.

Unlike these two CML data acquisition systems, TOP-RainCell undertook to install its own high frequency transmission/reception device: on the roof of the LA.M.E laboratory building was installed one of the two antennas, while the second one was installed in the telephony network of Telecel Faso to directly collect the performance data, as illustrated in Figure 1 below.

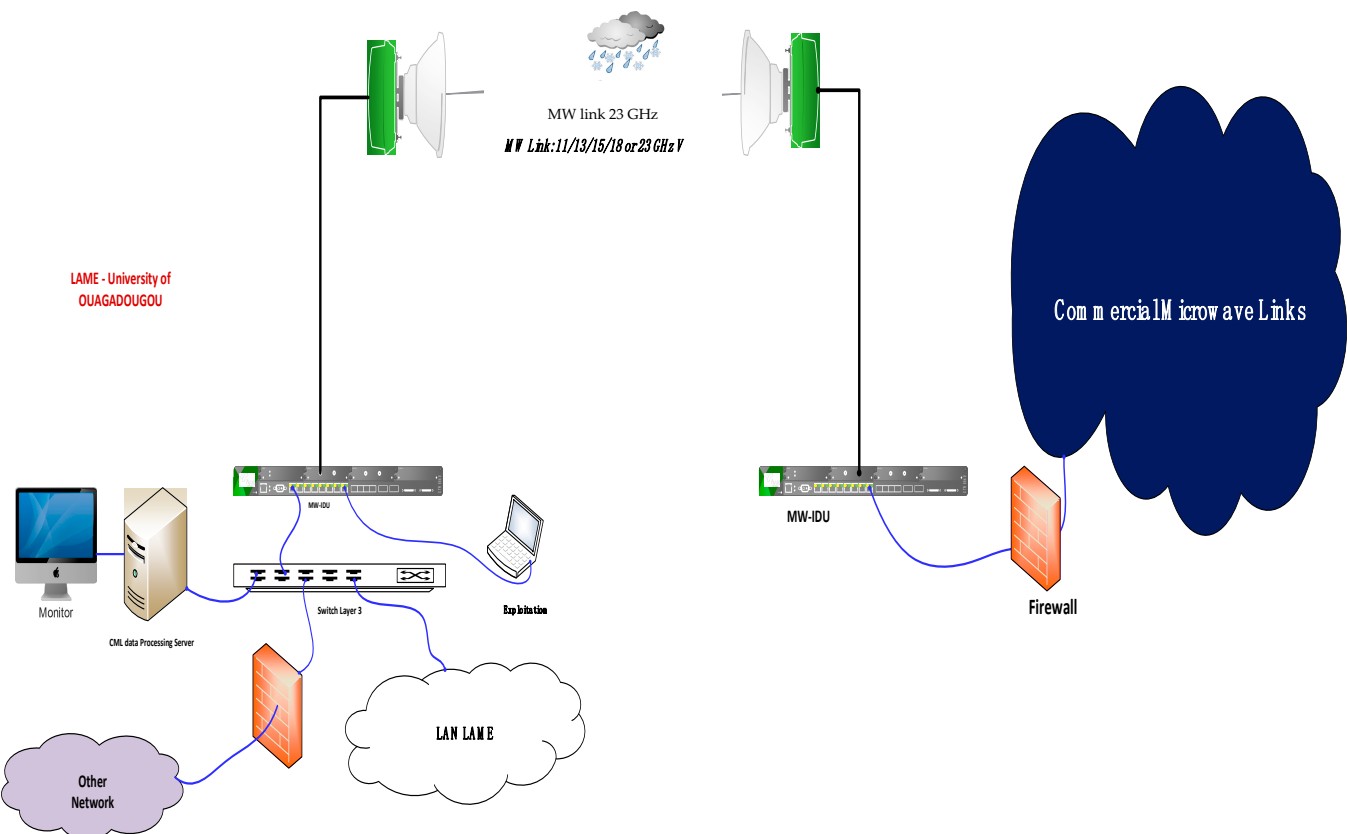

**Figure 1.** Schematic of Telecel Faso's CML power data acquisition system.

The CML data collection system presented in this paper is the first one to be installed in a research laboratory. This system records in real-time acquisition the power levels of the CMLs of the mobile phone operator Telecel Faso. These data will subsequently be used in future work, to quantify rainfall in Burkina Faso. In the following, we describe in detail the acquisition system and show data from this system.

## 2. Materials and Methods

### 2.1. Experimental Site

The experiment was conducted in Burkina Faso, a West African country with an area of 274,200 km$^2$, at latitude 12°30′00.00″ N and longitude 1°48′00.00″ W. Its capital, Ouagadagou, is located in the center of the country. Burkina Faso shares its borders with six neighboring countries: Mali at the northwest, Ivory Coast at the southwest, Ghana at the south, Togo and Benin at the southeast, and Niger at the northeast. The geographical position of Burkina Faso on the world map can be found in the reference [23].

As shown on the global rain-rate map [24], Burkina Faso is situated in the rain-rate zones, called the E, K, and N zones, in the ITU recommendations, that is to say, into the zones of 22 mm/h, 42 mm/h, and 95 mm/h. This particularity will be discussed in a next paper concerning CML data processing and results.

### 2.2. Acquisition System

The coverage of the territory by the network of the telephone operator Telecel Faso is dense. The level of density of network coverage is distributed according to the cities. It is therefore very appropriate to implement the above technology throughout the territory.

The first experiment was conducted on a single link, the Korsimoro–Kaya microwave link, in 2012. It was a CML link of 29 km long, with a transmission/reception frequency of 7 GHz, and whose antennas have horizontal polarization. As of 2018, thanks to the support

of Fonds National de la Recherche et de l'Innovation pour le Devéloppement (FONRID), the experimental zone was extended to the central region, i.e., the city of Ouagadougou and its region. On the one hand, from 2019 to 2021, the challenge was to achieve greater coverage of the national territory, and, on the other hand, TOP-RainCell wanted to record the CML data, with a time step of one minute, and automatically process these data to derive the rainfall amounts. Note that all the Telecel Faso's base stations might be "queried" directly and automatically by our data acquisition system.

For this purpose, a metadata file was built with the help of Telecel Faso since 2019 and regularly updated. This metadata file contained the identifiers, lengths, frequencies, and polarizations of the different links, as well as the latitudes and longitudes of the pylons.

The experimental setup was installed at the LA.M.E (Laboratoire de Matériaux et d'Environnement) of the Joseph KI-ZERBO University, in Ouagadougou in Burkina Faso. Two antennas of 0.3 m diameter with a minimum gain of 36.6 dBi were installed for the interconnection of the experimental setup installed at LA.M.E with the transmission center of Telecel Faso, as shown in Figure 1.

The installed device includes an indoor unit (IDU), which has the role of modulator/demodulator of the signals, and an outdoor unit (ODU), which transmits and receives the microwave signals. The link design was performed using pathloss 5.0 [25] and ITU-R rain Zone K, 66.26 mm/h inputs [26]. The inputs are provided below:

- calculation method: ITU-R P.530,
- ITU-R raine Zone K, (66.26 mm/h UIT-R P.837-7),
- average temperature °C: 27,
- water vapor density (g/m$^3$): 3.2,
- atmospheric pressure (Pa): 1013,
- frequency band: 23 GHz,
- channel spacing: 28 MHz (23 GHz-ITU-RF_737Ann3-CEPT13-02),
- modulation: ACM (mask of modulation 128 QAM, max 1024 QAM),
- antenna size: 1 Ft/0.3 m,
- radio type: ODU600v2,
- availability objective (%): 99.999.

The fading margin was 52.83 dB. The link throughput was 200 Mbps/128 QAM. A firewall was used to secure the system. Figure 2 and Table 1 provide, respectively, the microwave link in line of sight and the link budget details which were produced by Aviat Networks, Inc., Austin, TX, USA.

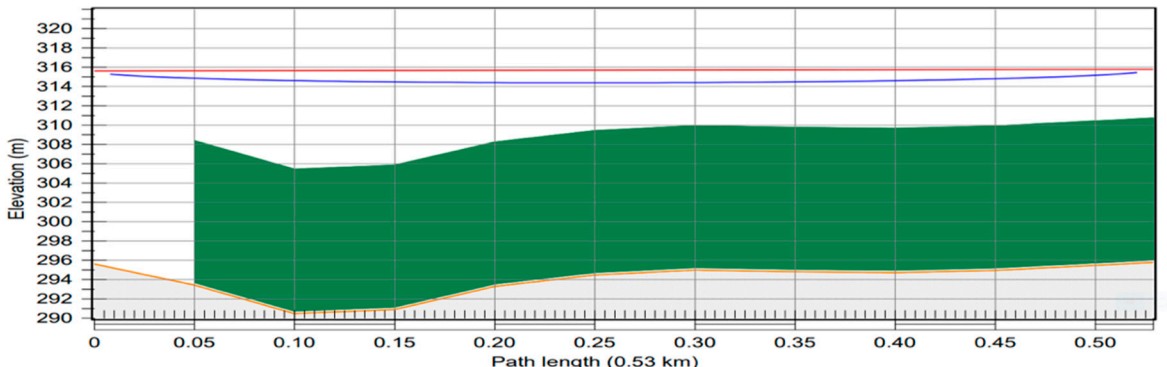

**Figure 2.** Microwave link in line of sight (Aviat Network, Inc. (July 2017)).

The implementation of this link required the signing of a partnership between LA.ME and Telecel Faso, and the assignation of a license by the "Autorité de Régulation des Communications Electroniques et des Poste (ARCEP)" of Burkina Faso, in order to use radio frequencies, as well as the transmitting and receiving devices.

**Table 1.** Link budget detail report (Aviat Network, Inc. (July 2017)).

| F = 23,000.00 MHz K = 1.33% F1 = 100.0, 60.0 | | |
|---|---|---|
| | **Telecel Universit** | **LA.M.E** |
| Latitude | 12 22 33.10 N | 12 22 43.90 N |
| Longitude | 001 30 05.53 W | 001 29 51.90 W |
| True azimuth (o) | 51.13 | 231.13 |
| Vertical angle (o) | 0.02 | −0.02 |
| Elevation (m) | 295.60 | 295.77 |
| Antenna model | VHLP1-23 (TR) | VHLP1-23 (TR) |
| Antenna gain (dBi) | 35.30 | 35.30 |
| Antenna height (m) | 20.0 | 20.0 |
| TX loss (dB) | 0.00 | 0.00 |
| RX loss (dB) | 0.00 | 0.00 |
| Configuration | 1 + 0 | 1 + 0 |
| Radio model | CRTE6hp18_28M 32Q 112Mb | CRTE6hp18_28M 32Q 112Mb |
| TX power (dBm) | 20.50 | 20.50 |
| EIRP (dBm) | 55.80 | 55.80 |
| Receive signal (dBm) | −23.17 | −23.17 |
| Thermal fade margin (dB) | 52.83 | 52.83 |
| Effective fade margin (dB) | 51.48 | 51.48 |
| Annual 2 way multipath availability (%) | 100.00000 | |
| Annual 2 way multipath unavailability (sec) | 0.00 | |
| Annual rain availability (%) | 100.00000 | |
| Annual rain + multipath availability (%) | 100.00000 | |
| Multipath fading method—Vigants—Barnett Rain fading method—Rec.ITU P 530-8/13 R(837-5) | | |

Figure 3 shows the acquisition system installed. It has been operational since July 2019 and allows access to more than 1000 CMLs of the Telecel Faso's network throughout Burkina Faso.

These CMLs are transmitting in the 6 GHz and 15 GHz frequency bands. The latter is used only in urban regions. One may note that attenuation increased with frequency [27]. However, this aspect will not be investigated in detail in this article.

The acquisition system comprises CMLs located in seven of the thirteen regions of Burkina Faso: « Le Nord», « Le Centre Nord», « Le Centre», « Le Centre Ouest», « La Boucle du Mouhoun», « Les Hauts Bassins », and « Les Cascades » (Figure 3). However, these regions are not covered in their entirety, except for the region « Le Centre». The CMLs concerned are those currently available for the cities of Ouagadougou, Bobo-Dioulasso, Banfora, Koudougou, Ouahigouya, and Kaya, plus those on the Telecel Faso's backbones.

Mobile phone operator Telecel Faso uses two types of links: urban connections and intercity connections.

Urban links are used within cities such as Ouagadougou, Bobo-Dioulasso, Banfora, Koudougou, Ouahigouya, Kaya, etc. Those links are short distance ones—from a few hundred meters to 10 km—transmitting on frequencies in the range between 10 GHz and 15 GHz.

Intercity connections are long-distance links—between 10 km and 50 km—transmitting on frequencies in the range between 6 GHz and 9 GHz. They are used to interconnect major cities of Burkina Faso. These include CMLs located on the axes Ouagadougou–Sabou–Boromo–Houndé–Bobo, Dioulasso–Banfora, Ouagadougou–Koudougou, Ouagadougou–Ziniaré–Kaya, and Ouagadougou–Gourcy–Ouahigouya. For all links, the antennas all

have a linear polarization which can be horizontal or vertical. It should be noted that horizontally polarized links are more sensitive to rain attenuation than vertically polarized ones [27].

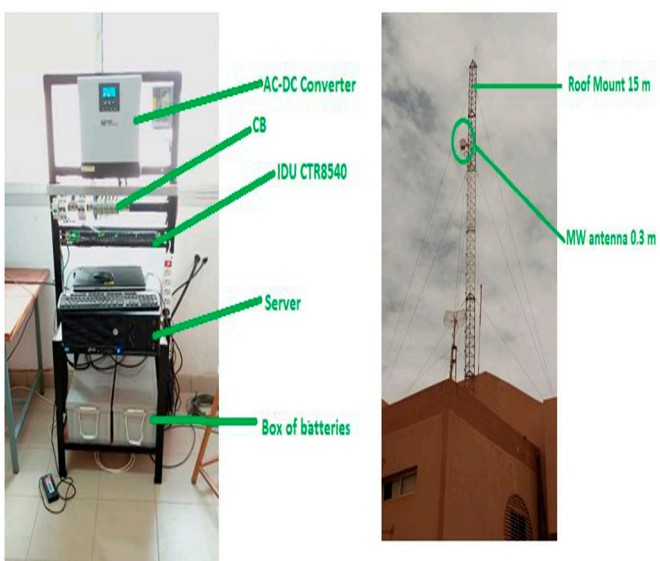

**Figure 3.** Acquisition system installed at the LA.M.E.

Figure 4 shows an overview of all the CMLs of the Telecel Faso's network in Burkina Faso connected to the data acquisition system (a), as well as the CML distribution in the cities of Ouagadougou (b), Bobo-Dioulasso (c), and Koudougou (d).

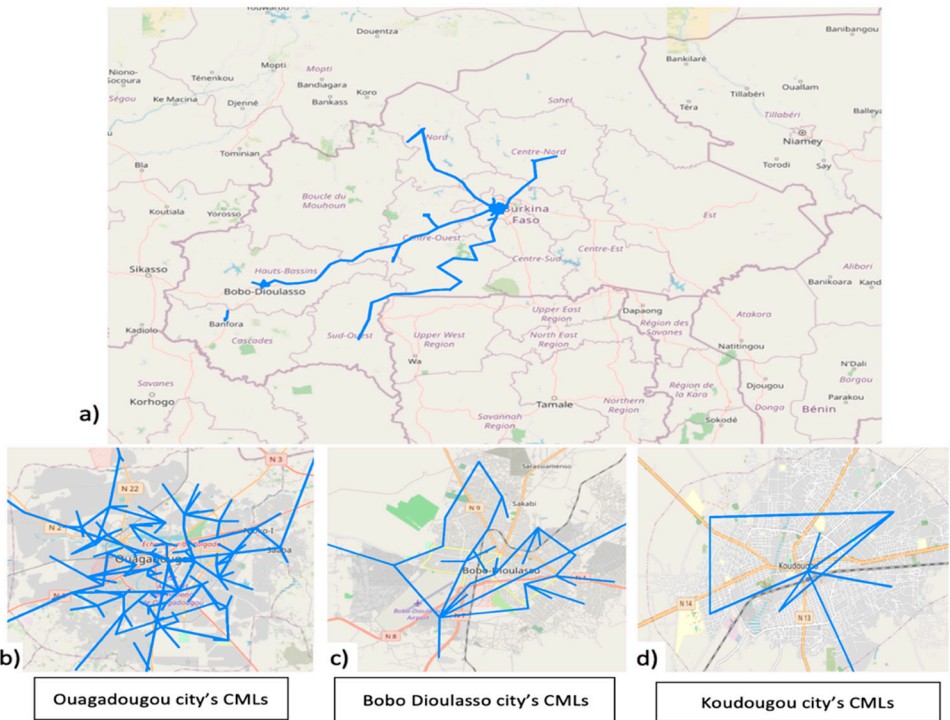

**Figure 4.** Map of the CMLs of the Telecel Faso's network in Burkina Faso (**a**) connected to the acquisition system, and zoom for the cities of Ouagadougou (**b**), Bobo-Dioulasso (**c**), and Koudougou (**d**).

*2.3. Data Acquisition Software*

The community of the microwave link's users for the quantification of precipitation has proposed several approaches and each one has demonstrated the possibility of using the telephone links to estimate the amount of rain fall:

○ https://github.com/fenclmar/Rcmlrain, developed under the R language, accessed on 5 February 2022,

○ https://github.com/cchwala/pySNMPdaq, using python language, accessed on 5 February 2022,

○ https://github.com/pycomlink/pycomlink, developed in a python environment, accessed on 5 February 2022,

○ https://github.com/overeem11/RAINLINK, using R language, accessed on 5 February 2022.

It needs to be noted that Rcmlrain, pycomlink, and RAINLINK do not carry out data acquisition. Only pySNMPdaq performs data acquisition. As long as only a few links were involved, in the case in 2019, there was no real question of choosing a method.

However, the ambition of TOP-RainCell is to work, in real time, on hundreds of links, even several thousands in the years to come.

TOP-RainCell has been engaged in a partnership with the University of Augsburg and the Karlsruhe Institute of Technology (KIT) in Germany for over seven years now. Code in Python has been developed there [10,18]. Therefore, it seemed judicious to choose the real-time SNMP-based CML data acquisition software, written in Python, pySNMPdaq. An adaption to the conditions for running it at Telecel Faso's in Burkina Faso had to be carried out, however.

*2.4. Description of Power Levels Collection Method*

For the acquisition of the CMLs' power level, we made use of an adjusted version of the Python software pySNMPdaq [22]. The SNMP V2 protocol has been used to remotely access the different CMLs of the Telecel Faso's network. Each CML link is accessible by the acquisition system via its IP address. An SNMP V2 request is defined by the IP address of the target base station and the SNMP object identifiers (OID) which are defined in the Management Information Base (MIB). Each OID identifies the object that can be recovered from a given CML, e.g., the transmitted power level denoted Tx, the received power level denoted Rx, the signal to noise ratio, or the hardware temperature. The OIDs for retrieving Tx and Rx typically differ, depending on the manufacturer and specific hardware model [22]. Two kinds of Aviat series IDUs, CTR 8312 and CTR 8540, are those interconnected to the acquisition system.

The data acquisition (DAQ) server requests the power levels of the transmitted and received signals of each link, and instantly stores these data in a file (Figure 5). The data acquisition program runs in a background loop, as a daemon program, and collects the Tx and Rx power data from the various CMLs at the rate of one minute.

To ensure constant synchronization of simultaneous SNMP V2 requests, the program is subdivided into three sub-processes (Figure 5). First, there is the subprocess that continuously triggers data request events with a time resolution of one minute. This task continues in the background. Next, a data acquisition sub-process is started for each CML. This sub-process manages the SNMP V2 communication with the CMLs. Finally, it follows the process that receives all data from all links via a queue and stores them in files. The architecture thus adopted is illustrated in Figure 5.

The SNMP requests are launched from the acquisition server via the microwave link that interconnects the LA.M.E's local network to Telecel Faso's network. These requests are launched at the same time for all the CMLs the system has access to.

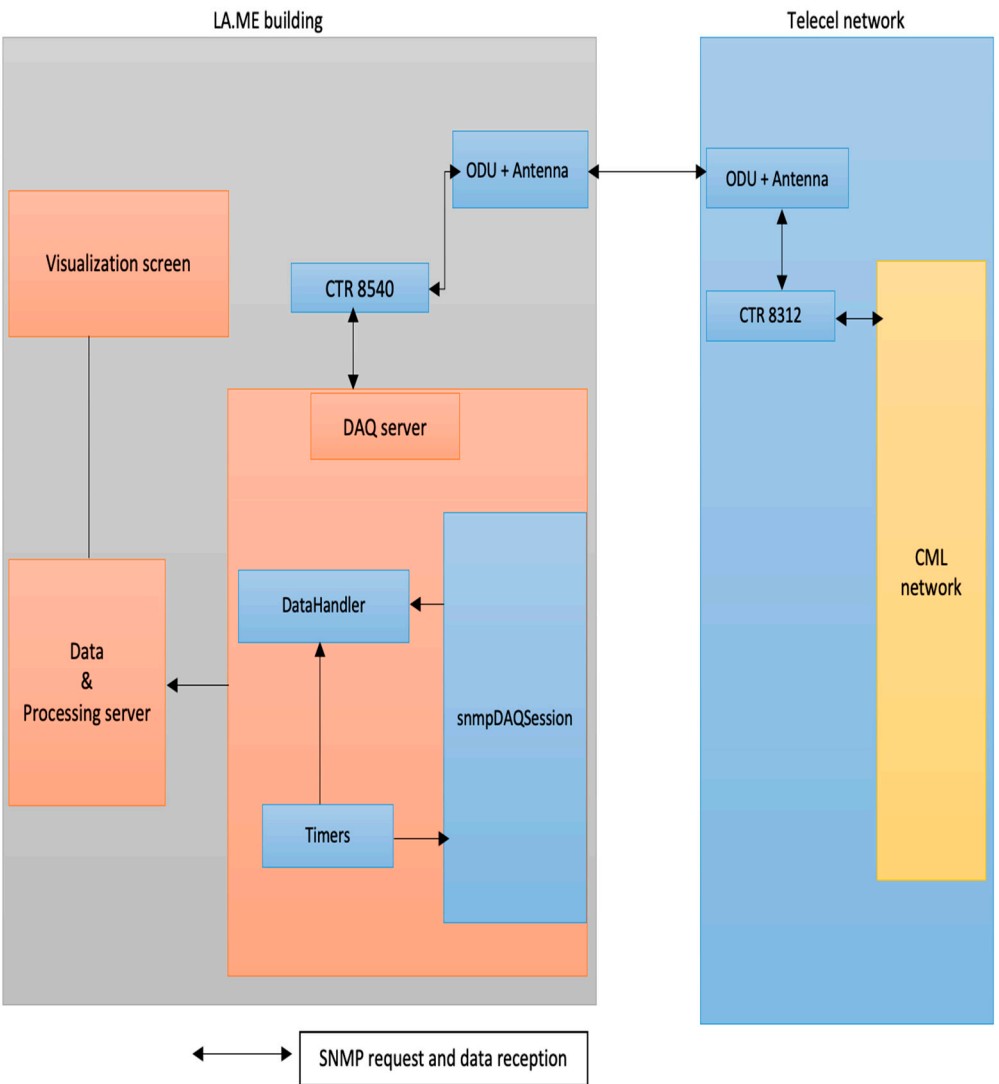

**Figure 5.** Software structure of the data acquisition system.

For better management of the data, a folder is created for each CML as soon as it is integrated into the system. Inside this folder, one data file is created per hour for the considered link. This later data file records the Tx and Rx power levels of the CML, with a temporal resolution of one minute. This storage method allows for efficient data management. Indeed, it is possible to easily access and process the data of a given CML over a given time range. It is also possible to load and process all the data of one or several CMLs.

One may note that this acquisition system guarantees the integrity of all data of the mobile telephony operator Telecel Faso.

## 3. Overview of Data Collected by the Acquisition System

Figure 6 shows a screenshot of the collected data. Columns one to six provide, respectively, the date, hour, name of the CML link, round trip time of the SNMP request (in millisecond), received signal level (in dBm), as well as the transmitted signal level (in dBm).

```
2020-06-01| 00:00:01,| BOROMO—OUAHABOU | , 1.011     ,|     −38.2,|     29.9 |
2020-06-01| 00:01:00,| BOROMO—OUAHABOU | , 0.0114    ,|     −38.3,|     30.1 |
2020-06-01| 00:02:01,| BOROMO—OUAHABOU | , 1.0112    ,|     −38.0,|     30.1 |
2020-06-01| 00:03:00,| BOROMO—OUAHABOU | , 0.0105    ,|     −38.0,|     29.9 |
2020-06-01| 00:04:00,| BOROMO—OUAHABOU | , 0.0546    ,|     −38.2,|     30.1 |
2020-06-01| 00:05:00,| BOROMO—OUAHABOU | , 0.0113    ,|     −38.0,|     30.1 |
2020-06-01| 00:06:01,| BOROMO—OUAHABOU | , 1.011     ,|     −38.0,|     29.9 |
2020-06-01| 00:07:00,| BOROMO—OUAHABOU | , 0.0114    ,|     −38.1,|     29.9 |
2020-06-01| 00:08:00,| BOROMO—OUAHABOU | , 0.0103    ,|     −38.0,|     30.1 |
2020-06-01| 00:09:00,| BOROMO—OUAHABOU | , 0.0108    ,|     −38.0,|     29.9 |
2020-06-01| 00:10:01,| BOROMO—OUAHABOU | , 1.0112    ,|     −38.1,|     29.9 |
2020-06-01| 00:11:00,| BOROMO—OUAHABOU | , 0.0103    ,|     −38.1,|     29.9 |
2020-06-01| 00:12:01,| BOROMO—OUAHABOU | , 1.0155    ,|     −38.1,|     29.9 |
2020-06-01| 00:13:00,| BOROMO—OUAHABOU | , 0.0108    ,|     −38.3,|     29.9 |
2020-06-01| 00:14:01,| BOROMO—OUAHABOU | , 1.0113    ,|     −38.1,|     29.9 |
2020-06-01| 00:15:00,| BOROMO—OUAHABOU | , 0.012     ,|     −38.3,|     29.9 |
2020-06-01| 00:16:01,| BOROMO—OUAHABOU | , 1.0114    ,|     −38.5,|     29.9 |
2020-06-01| 00:17:00,| BOROMO—OUAHABOU | , 0.0103    ,|     −38.2,|     30.1 |
2020-06-01| 00:18:02,| BOROMO—OUAHABOU | , 2.0122    ,|     −38.2,|     30.1 |
2020-06-01| 00:19:00,| BOROMO—OUAHABOU | , 0.0104    ,|     −38.1,|     29.9 |
2020-06-01| 00:20:01,| BOROMO—OUAHABOU | , 1.0138    ,|     −38.3,|     29.9 |
2020-06-01| 00:21:00,| BOROMO—OUAHABOU | , 0.0105    ,|     −38.3,|     29.9 |
2020-06-01| 00:22:00,| BOROMO—OUAHABOU | , 0.0111    ,|     −38.3,|     30.1 |
2020-06-01| 00:23:00,| BOROMO—OUAHABOU | , 0.0112    ,|     −38.3,|     29.9 |
2020-06-01| 00:24:01,| BOROMO—OUAHABOU | , 1.0119    ,|     −38.3,|     29.9 |
2020-06-01| 00:25:00,| BOROMO—OUAHABOU | , 0.0109    ,|     −38.3,|     29.9 |
2020-06-01| 00:26:01,| BOROMO—OUAHABOU | , 1.0116    ,|     −38.2,|     29.9 |
2020-06-01| 00:27:00,| BOROMO—OUAHABOU | , 0.0154    ,|     −38.4,|     29.9 |
2020-06-01| 00:28:01,| BOROMO—OUAHABOU | , 1.0122    ,|     −38.5,|     29.9 |
2020-06-01| 00:29:00,| BOROMO—OUAHABOU | , 0.0111    ,|     −38.5,|     30.1 |
2020-06-01| 00:30:01,| BOROMO—OUAHABOU | , 1.0145    ,|     −38.4,|     29.9 |
2020-06-01| 00:31:00,| BOROMO—OUAHABOU | , 0.0115    ,|     −38.4,|     29.9 |
```

**Figure 6.** Screenshot of collected data.

Figure 7 shows an example of the collected data for one CML between July and December 2019. As mentioned before, the system allows for each link to obtain the transmitted power (Tx [dBm]) and the received one (Rx [dBm]), with a time step of one minute. The attenuation of the CML signal is obtained by taking the difference between the transmitted and received powers (Tx − Rx) [16]. The transmitted power is constant when the automatic transmit power control (ATPC) is not activated, as is the case in the example shown. The received power, and thus also the attenuation, changes according to weather conditions over the link. We can note from this figure that the attenuation of the link varies over time, as expected. The other links provided similar data.

Figure 8 shows the attenuation time series of two CMLs integrated to the acquisition system and the rainfall data from two rain gauges of the Agence Nationale de la Météorologie (ANAM) of Burkina Faso, for the rainfall events of 31 July 2019, and 1 August 2019. The data from the rain gauges are used here as references for the CML data validation. The two rain gauges used in this experiment are installed in Ouagadougou, Burkina Faso, at the SOMGANDE and KAMBOINSSIN sites and record rainfall intensities at a 15 min time step.

The telecommunications commercial microwave links considered are KAMBOINSSIN-KAONGHIN and CNTS-PASPANGA. The KAMBOINSSIN-KAONGHIN link is 5.68 km long, transmitting at 13.031 GHz, and has vertically polarized antennas. The CNTS-PASPANGA link is 0.96 km long, transmitting at 12.87 GHz with vertical polarization.

These two links have been chosen due to their geographical proximity to the two rain gauges of KAMBOINSSIN and PASPANGA, respectively.

Since the CML data were collected with a time resolution of one minute and in order to measure the dynamic coherence with the rain gauge data, these data have been resampled at a time step of 15 min.

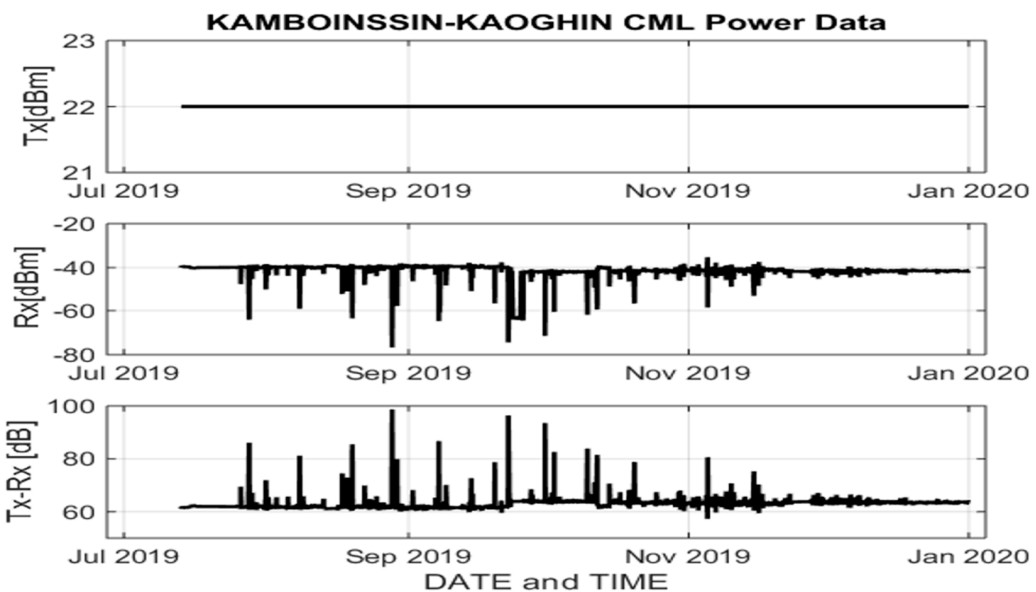

**Figure 7.** Transmitted power, received power, and attenuation of the KAMBOINSSIN-KAOGHIN link.

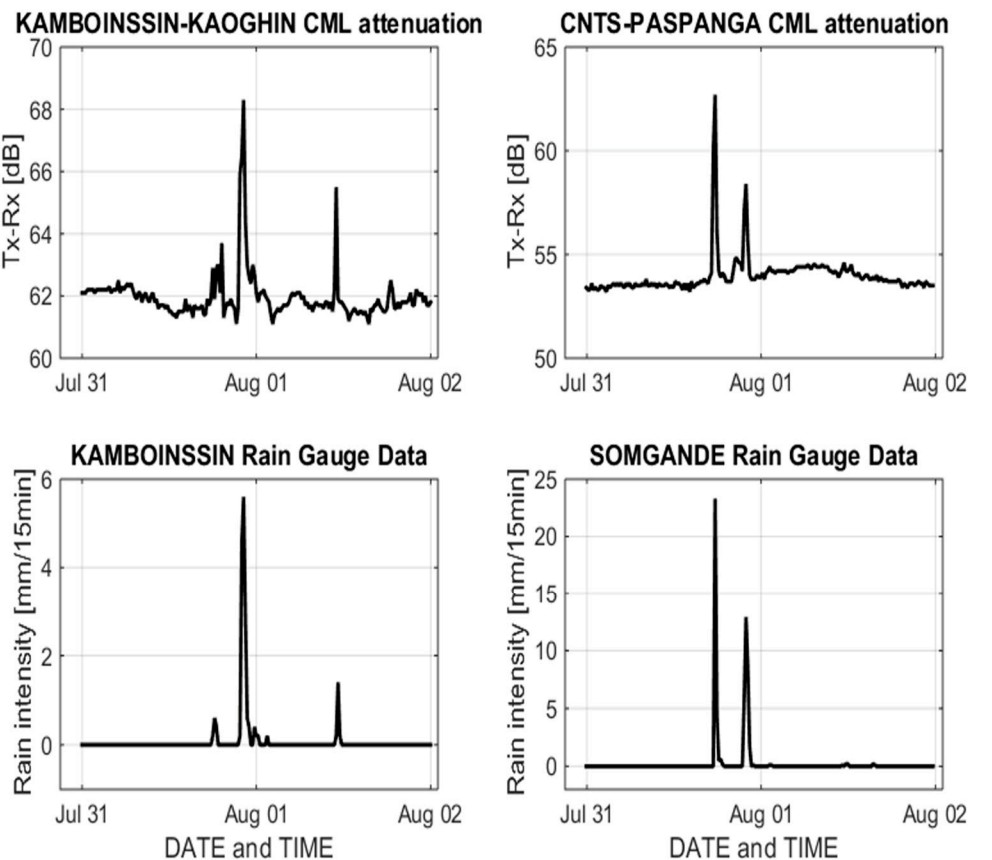

**Figure 8.** CMLs' attenuations data and ANAM rain gauge data for the 31 July 2019 and 1 August 2019 rain events.

In Figure 8, we can observe a very good coherence between the rain gauge data and the CML data. Indeed, each rainy event is characterized by the presence of a peak in the CML and/or gauge data. We can see on this figure that the main peaks of the CML and gauge

data are positioned at the same locations. Thus, all the rainfall events observed by the rain gauges are also visible on our links, confirming the quality of our CML data. We can therefore confirm that the use of CML data is an alternative, innovative, and inexpensive solution for rainfall quantification.

### 4. Protection and Security of the Acquisition System

The follow up and the maintenance of the experimental setup were also two very important questions to which a great interest has been paid. This concerns the transmission/reception devices and the electrical energy.

Regarding the high frequency (HF) transmission/reception device, there were three aspects:

- Lightning protection: on the building a lightning rod was considered essential to be installed. The choice was made to install a lightning conductor with a radius of 30 m all around the LA.M.E laboratory building.
- Verification and reinforcement, or even extension of the mast, installed in 2018, to ensure a good performance of the high frequency transmission/reception system even in strong winds.
- Verification of the high frequency transmission/reception device installed on the ground, in the LA.M.E laboratory.

Secondly, the other component of the follow up and the maintenance was the verification and monitoring of the secondary electrical power system. Indeed, it should be noted that power cuts are frequent in Burkina Faso, particularly in the city of Ouagadougou. These sudden interruptions, sometimes followed or preceded by sharp rises or falls in the level of the supply voltage, are very harmful to equipment, even fatal. It was therefore necessary to install a second energy source to ensure the continuous supply of the system. Therefore, a Photo Voltaic (PV) electrical generator as a secondary source was dimensioned and installed. This secondary source should work permanently in both dry and rainy seasons, thanks to the backup batteries of the PV system. On the other hand, regarding this point, it is necessary that the system collects the performance data of available power throughout the year, in dry or wet seasons, to be able to monitor that no interruption of energy has occurred.

### 5. Conclusions

This paper presents a real-time power level data acquisition system for CMLs. The system, installed at the LA.M.E of Joseph KI-ZERBO University, uses SNMP V2 protocol, to collect transmitted and received power data from Telecel Faso's CMLs connected to the system. The number of CMLs accessible to the system has progressively increased from 8 links, in 2019, to nearly 1000 links currently, covering the capital Ouagadougou and the main cities of Burkina Faso (Bobo Dioulasso, Ouahigouya, Koudougou, and Kaya) as well as the axes connecting Ouagadougou to these cities. The attenuation data collected by the system show a good correlation with the data from nearby ANAM rain gauges, showing that the system is working well.

Future works will focus on increasing the number of CMLs accessible by the system. Another important step will be to process the collected data in order to estimate rainfall amounts and to derive a rainfall map with high temporal and spatial resolution in Burkina Faso. Short-time rainfall prediction will also be investigated.

**Author Contributions:** Conceptualization, A.D., S.R.S., M.S. and F.Z.; methodology, M.D., S.R.S., M.S., C.C. and H.K.; software, M.D., S.R.S., M.S. and C.C.; validation, S.R.S., M.S., C.C., H.K. and F.Z.; formal analysis, H.K. and F.Z.; investigation, A.D., S.R.S., M.S., I.G., C.C., H.K. and F.Z.; resources, M.D., I.G., W.Y.S.B.O. and N.K.; data curation, M.D., S.R.S., M.S., C.C. and F.Z.; writing—original draft preparation, M.D. and W.Y.S.B.O.; writing—review and editing, M.D., S.R.S., M.S., A.D., W.Y.S.B.O., C.C., H.K. and F.Z.; visualization, M.D. and C.C.; supervision, F.Z.; project administration, H.K. and

F.Z.; funding acquisition, H.K. and F.Z. All authors have read and agreed to the published version of the manuscript.

**Funding:** Thanks to "Fonds National de la Recherche et de l'Innovation pour le Développement" (FONRID) of Burkina Faso for the funding which allowed the experimental zone to cover Ouagadougou and its region, and UNDP who funded TOP-RainCell during the 2020 monsoon season. AgRAIN project funds of one the authors since September 2019. The APC of this article was supported by "Université Joseph KI-ZERBO", UJKZ.

**Acknowledgments:** This work was carried out thanks to several partners. First, we would like to thank Telecel Faso for access to its network to collect CMLs' data and ARCEP for the license to operate the CMLs. Thanks to "Fonds National de la Recherche et de l'Innovation pour le Développement" (FONRID) of Burkina Faso for the funding which allowed the experimental zone to cover Ouagadougou and its region. We also thank the project AgRAIN, funded by the German Ministry of Research and Education (BMBF). We cannot finish without thanking the UNDP who funded TOP-RainCell during the 2020 monsoon season and Université Joseph KI-ZERBO for the payment of the publication fees.

**Conflicts of Interest:** The authors declare no conflict of interest.

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
