# Peer review of "Towards Innovative Solutions for Monitoring Precipitation in Poorly Instrumented Regions: Real-Time System for Collecting Power Levels of Microwave Links of Mobile Phone Operators for Rainfall Quantification in Burkina Faso"

_asi, doi:10.3390/asi6010004_

Round 1

Reviewer 1 Report

The paper is well written and it is a great job.

Reviewer 2 Report

Dear Respected Authors,

I have found your paper interesting and I have red it with pleasure. I concern it as a kind of application-oriented work with lot of information about system-level configuration. You have used the tool, which is a network of CMLs to obtain high-level data and analyze it. From this point of view it is interesting as a way to solve a problem with rainfall evaluation.

On the other hand, the scientific part of the paper seams to be covered by the application oriented information. The paper is interesting as a story but the scientific value-added should be emphasized. There are my remarks below:

1. I propose to add some additional information about microwave attenuation in atmosphere considered for various frequency bands. You have used several bands, what is the advantage when you compare these data.

2. Please comment the issue of various polarization -  what would be the advantage if you have the opportunity of use different polarizations of EM wave.

3. What about aggregation of data consisting information of wind speed ? Atmospheric pressure ? Would it be a kind of development direction for your system ?

4. What about aggregation of other information from other sources, for example, short distance recordings of signal level from wireless internet providers ? Maybe a kind of free-space optical links ?

5. In paper there are several typing "inconsistencies" for example: 

- LAME versus LA.ME,

- TOP-RAINCELL versus TOP RainCell versus Top RainCell,

Best regards

Reviewer 3 Report

Dear Scientific Editor, Dear Authors,

The submitted paper "Towards innovative solutions for monitoring precipitation in 1 poorly instrumented regions: Real-time system for collecting 2 power levels of microwave links of mobile phone operators for 3 rainfall quantification in Burkina Faso" is a very interesting reading and a useful summary of local conditions about a relatively less-known region. As a reviewer, I find extremely fruitful the presented co-operation of various domains including meteorological, telecom operator, agricultural and university experts from different countries. My general opinion is very positive, I definitively recommend the paper for publishing as it is an important contribution on how a commercial microwave access network can be used to verify existing empirical and non-terrestrial (satellite or radar based) microwave data collection methods. Special value will be the locally collected long term data specific for the region discussed.

Most of my comments are just recommendations, where I leave the decision for the Authors and the Scientific Editor to accept these comments or not. None of these recommendations or comments are critical. However, there are a few editorial and technical mistakes in the paper, that I definitively ask to correct before publishing, to improve the quality of the final paper.

General recommendations and comments:

The paper is quite short. It would be very nice to add a map in the introduction part, and some explanation where Burkina Faso is. Especially, into which ITU rain-zone Burkina Faso belongs to, e.g. K-zone of 42 mm/h or multiple rain-zones. Line 177 mentions ‘region K’ but referred link is not about rain zones (I may be wrong, but the given URL is about interference). Please see for example (just for illustration) the rain zone map Figure 2 of ITU-R PN.837.1 (or newer ITU recommendation version of same ITU-R 837…),

or Fig.17-19 in Availability and Fade Margin Calculations for 5G Microwave and Millimeter-Wave Anyhaul Links, MDPI Applied Sciences, Vol.9, Issue.23, 5240, Dec. 2019. DOI:10.3390/app9235240. Readers believe that Burkina Faso is in K-zone, but naturally the resolution of ITU maps is not sufficient to show where Burkina Faso is exactly.

Similarly, a copy of Path Loss link design would be a nice illustration about the 23 GHz Vertical link, having 36.6 dBi antennas. A simple link budget would show how 99.9999% is derived in Line 178. This would make the paper a bit longer, but the paper is short, there is space I suppose. For link budget / link calculation please see the paper above or for example ‘Throughput Estimation of K-zone Gbps Radio Links Operating in the E-band’ in Informacije MIDEM, Journal of Microelectronics, Electronic Components and Materials, Vol.52, No.1, pp.29-39, ISSN: 2232-6979, 2022. DOI: 10.33180/InfMIDEM2022.104. Most probably Path Loss gives some tabular output how received signal level is calculated. This would help the understanding of link design of Figure 1, including local rainfall conditions (reader supposes it is 42 mm/h in 0.01% of the time.)

Editorial corrections required:

From Line 100: reference [23] follows reference [16] in Line 87. Reference [22] comes after [23]. I request to follow MDPI style, and order all references in a consecutive order, as they appear in the text after each other.

Line 90-91. The Roman numbers skip iii in the list. Either iv corrected to iii or a brand new point added. As point iii, I would proudly mention that the developed system is fully automatic, and it does not require experienced service personnel to operate it, (as it is needed for the radar type tests as mentioned in Line 72). I leave the decision for the Authors how to correct.

Line 131. (2015) written twice, it is not needed in brackets.

Lines 149-150 Please keep titles together with paragraph, do not break titles and their chapters over a page.

Link 195 and 214: the same sentence is repeated.

Line 217: better resolution of the maps would be welcomed if available.

Line 219: a)) has two closing brackets.

Line 264, Figure 4. The figure has low resolution, it is difficult to read the texts inside the boxes.

Line 267: LA.ME is written with point everywhere. Here LA.ME is LAME. Also some figures use LAME.

Line 376: maybe conflict of interest is missing. I leave this decision for the Scientific Editor, to follow MDPI ASI style.

Line 372: Funding may be in a separate chapter of Fundings, I leave this decision for the Scientific Editor, to follow MDPI ASI style.

Question: I am very sorry for my weak knowledge of the country’s geography, but is PAPANGA in Line 306 the same city as PASPANGA in Figure 7? What is the proper name of the cite: Papanga or PaSpanga?

Line 427, reference 21. Please add ITU-T recommendation number and title too. The provided link does not match, Series K is about interference and not rain…

Technical mistakes, where correction or some explanation required:

Line 178. Correct spellings are fade margin or fading margin. Please see e.g. A.Hilt, Microwave Hop-Length and Availability Targets for the 5G Mobile Backhaul, IEEE 42nd Telecommunications and Signal Processing Conference, Hungary, July 2019, DOI: 10.1109/TSP.2019.8768870, or paper mentioned above.

Line 179: ‘channel spacing’ shall not use title case. It should be also mentioned if the 128-QAM link uses adaptive modulation or not. Is the link switching automatically down to 64 or 32-QAM in case of strong rain or interference? If yes, then different Tx and Rx threshold levels may characterize the link. Readers/Reviewers may not know Aviat microwave radios in such detail.

Line 280: round trip time is shown in the Figure, but its unit is not specified. Reviewer believes it is in milliseconds, but not 100% sure. In the same line it would be nice to say if signal levels are in dBm.

Line 290-291. Reviewer believes it is a mistake. Tx power is constant when ATPC is not activated. Please check, ATPC is for Automatic Transmit Power Control. AGC is usually mentioned for the receivers, where the received signal level is varying and the amplifier equalizes this to a constant level. But the reviewer is not 100% sure, I do not know Aviat CML in such details. Please check.

Line 296, Figure 6: unit in y-axes are incorrect. Please correct. Rx and Tx levels are power levels, with a physical unit of dBm. In dBm unit, the reference is 1 mW, so in brackets it must be [dBm]. Tx-Rx is attenuation, that is unitless. Here the unit [dB] is correct. Attenuation, Gain and Loss are unitless, they are just a ratio of output compared to the input. (Research paper shall not be published in MDPI with mistakes in physical dimensions!)

Line 322, figure 7, Rain Gauge Data. On Axis y the unit should be mm/time. Intensity means mm/h or mm/15 minutes. As paper discusses 15 min resolution, most probably the unit is mm/15 minutes, but readers are not sure. Alone [mm] would mean rain amount, and not rain intensity.

Just a comment: knowing the frequency, polarization and the hop length of the links between KAMBOINSSIN-KAOGHIN and CNTS-PAPANGA, the measured attenuation could give estimation for the actual rain intensity of the days discussed. These could be compared to the measured data of the rain gauge. For calculation details pls. see papers mentioned above. Maybe such calculation cannot fit into the timeframe of this paper but definitively recommended for the future work.

In general, above can be fixed quickly. The paper is very interesting, it is a multi-disciplinary approach of several researchers from different fields. There is a strong need for local validation of ITU rain zones, rain intensities. The presented method is a very practical engineering approach with moderate investment, rather benefiting the exisiting links. Huge number of real telecom links are investigated in real time. I definitively recommend the paper for publishing in MDPI. One reason for this is that European/American or Asian readers have no practical knowledge of the local conditions in Burkina Faso. Another reason is the excellent approach of the local Mobile Network Operator to use its resources for the support of research and economy (agriculture) and publishing the results together with colleagues from Acadeimia. The paper is a bit short, I would bravely recommend to extend the paper with some more details e.g. Path Loss calculation of the 128 QAM data collecting link. 
Best regards, the reviewer (27-Oct-2022)
